# Biorefinery Approach Applied to the Production of Food Colourants and Biostimulants from *Oscillatoria* sp.

**DOI:** 10.3390/biology11091278

**Published:** 2022-08-28

**Authors:** Ainoa Morillas-España, Ruperto Bermejo, Roberto Abdala-Díaz, Ángela Ruiz, Tomás Lafarga, Gabriel Acién, José María Fernández-Sevilla

**Affiliations:** 1Department of Chemical Engineering, University of Almería, 04120 Almeria, Spain; 2CIESOL Solar Research Centre, Joint Centre University of Almería-CIEMAT, Ctra. Sacramento s/n, 04120 Almería, Spain; 3Department of Physical and Analytical Chemistry, Higher Engineering Polytechnic School of Linares, University of Jaén, 23700 Linares, Spain; 4Departamento de Ecología y Geología, Facultad de Ciencias, Instituto de Biotecnología y Desarrollo Azul (IBYDA), Universidad de Málaga, 29071 Málaga, Spain

**Keywords:** biorefinery, phycocyanin, biostimulants, microalgae, toxicity, bioeconomy

## Abstract

**Simple Summary:**

In this study, a biorefinery based on *Oscillatoria* sp. is developed to produce high-value compounds such as C-phycocyanin, used in food colourant applications, and biostimulants, used in agriculture-related applications. The results confirm that C-phycocyanin concentrations ranging from 22 to 106 mg/L produce colours similar to commercial products; moreover, the safety of the extracted C-phycocyanin was confirmed through toxicity tests. The leftover biomass was confirmed as a biostimulant, with the results confirming a relevant auxin-like positive effect. Finally, an economic analysis was conducted to evaluate different scenarios, with results confirming this as the best scenario from an economic standpoint.

**Abstract:**

In this study, a biorefinery based on *Oscillatoria* sp. is developed to produce high-value compounds such as C-phycocyanin, used in food colourant applications, and biostimulants, used in agriculture-related applications. First, the *Oscillatoria* biomass production was optimized at a pilot scale in an open raceway reactor, with biomass productivities equivalent to 52 t/ha·year being achieved using regular fertilizers as the nutrient source. The biomass produced contained 0.5% C-phycocyanins, 95% of which were obtained after freeze–thawing and extraction at pH 6.5 and ionic strength (FI) 100 mM, with a purity ratio of 0.71 achieved in the final extract. This purity ratio allows for use of the extract directly as a food colourant. Then, the extract’s colourant capacity on different beverages was evaluated. The results confirm that C-phycocyanin concentrations ranging from 22 to 106 mg/L produce colours similar to commercial products, thus avoiding the need for synthetic colourants. The colour remained stable for up to 12 days. Moreover, the safety of the extracted C-phycocyanin was confirmed through toxicity tests. The waste biomass was evaluated for use as a biostimulant, with the results confirming a relevant auxin-like positive effect. Finally, an economic analysis was conducted to evaluate different scenarios. The results confirm that the production of both C-phycocyanin and biostimulants is the best scenario from an economic standpoint. Therefore, the developed biomass processing scheme provides an opportunity to expand the range of commercial applications for microalgae-related processes.

## 1. Introduction

Microalgae, including cyanobacteria, are a potential source of bioproducts that can be used for applications in high-value markets, such as pharmaceuticals, cosmetics, nutraceuticals, and food; medium-value markets, such as feed, biomaterials, biostimulants, and biopesticides; and low-value markets, including biofuels and biofertilisers [1]. Despite their great potential, the annual production of microalgae is relatively small, not exceeding 40,000 t worldwide and lower than 3000 t in Europe [2]. Due to the current low production capacity, most of the biomass produced is destined for the nutraceutical and food sectors, with a small share being used in cosmetics and aquafeeds. The main reason is that, in these applications, the biomass value is sufficient to cover the current production costs; these range from 20–50 €·kg^−1^ for the biomass produced in closed photobioreactors and from 5–20 €·kg^−1^ for those produced in open raceways [1]. 

To promote the utilisation of microalgal biomass in other medium-value and low-value biomass sectors, two strategies have been suggested: (i) to reduce the current production costs and (ii) to develop biorefinery schemes able to provide a wide portfolio of products combining high, medium, and low-value products. To reduce the current production costs, new production technologies need to be developed and current production processes need to be scaled up; these changes could take place in the medium term. In contrast, the development of biorefinery schemes might be achieved in the short term. The biorefinery concept involves obtaining multiple products from the biomass, where medium- and low-value products are recovered after the high-value compounds—such as pigments, polyunsaturated fatty acids, vitamins, etc.—have been removed [3]. To develop such biorefinery schemes, it is necessary to identify the right products based on the composition of the biomass produced and the market price/demand for the target compounds. 

Pigments such as chlorophylls, carotenoids, and phycobiliproteins are valuable compounds that have traditionally been obtained from microalgae [4]. Of these, phycobiliproteins are particularly interesting given their various applications; they are currently used as fluorescent markers, nutraceuticals, and food colourants, among other things [5]. Regarding the food additives market, natural colourants are of great interest. Phycobiliproteins have been incorporated into different food matrices, although their use needs to be further supported by toxicity testing, which is, until now, an aspect generally overlooked. These valuable compounds are naturally produced by cyanobacteria, with strains such as *Spirulina* and *Oscillatoria* being especially well-suited for producing phycobiliproteins (C-PC) [5]. Some authors have claimed that the use of this protein in food and beverages is limited by their lack of stability to heat, light, and pH variations [6]. The biostimulants market has also received increased attention over the last decade due to a greater demand for sustainable agriculture, as well as the need to boost crop yields and reduce the carbon footprint from food production. The biostimulants market was estimated to be worth USD 2.6 billion in 2019 and is projected to reach USD 4.9 billion by 2025, at a CAGR of 11.24% over the forecast period [7]. Microalgae are already recognised as a source of valuable biostimulants for agriculture due to their amino acid and phytohormone contents acting as plant growth promoters [8]. Several microalgal strains have been proven to be natural producers of phytohormones, such as indole-3-acetic acid (IAA), cytokinins, gibberellins, abscisic acid (ABA), and jasmonic acid, as well as other substances with biostimulant activity [9]. Cyanobacteria, such as *Arthrospira* and *Oscillatoria,* can also produce IAA, cytokinins, and/or jasmonic acid [10]; their use in agriculture improves the plants’ ion uptake, antioxidant content, cellular metabolism, and leaf chlorophyll content as well as the soil’s nutrient availability and humidity [11,12].

For these reasons, *Oscillatoria* has been identified as a promising raw material for both food colourants and biostimulants. This work aimed to analyse the possibility of developing biorefinery schemes using *Oscillatoria* for the production of food-grade C-PC and biostimulants for use in agriculture. The nontoxic nature and the colouring properties of the C-PC extracts were evaluated along with the positive biostimulant effects of the leftovers from C-PC extraction. A technoeconomic analysis was carried out to identify the major bottlenecks in developing real industrial processes focused on these products.

## 2. Materials and Methods

### 2.1. Microalgae Production

The cyanobacterium *Oscillatoria* sp. was produced in seawater using a culture medium based on commercial fertilizers: 0.90 g·L^−1^ NaNO_3_, 0.25 g·L^−1^ MgSO_4_, 0.3 g·L^−1^ K_2_PO_4_, and 0.03 g·L^−1^ of Karentol^®^ (Kenogard, Spain). The photobioreactor used was a 12 m^3^ (80 m^2^) raceway reactor located inside a greenhouse at the SABANA Demonstration Plant (IFAPA, Almería, Spain). The biomass was produced in semicontinuous mode at a 0.3 day^−1^ dilution rate. The temperature, pH, and dissolved oxygen concentration of the culture were monitored online using 5083T and 5120 probes (Crison Instruments, Barcelona, Spain) connected to an MM44 control-transmitter unit (Crison Instruments, Barcelona, Spain) and Labview data acquisition software (National Instruments, Austin, TX, USA). The pH was controlled at 8.0 by on-demand injection of CO_2_ whereas the dissolved oxygen was kept below 250%Sat by on-demand aeration. The volume of the reactor was kept constant by adding freshwater (evaporation) and the pH was controlled by CO_2_ injection.

### 2.2. Extraction and Characterization of Phycocyanin

C-phycocyanin was extracted from the freeze-dried biomass by solubilisation in phosphate buffer (5 mM, pH 6.5) at a ratio of 1:20 (*w*/*v*). The solution was stirred for 5 min at 2000× g with a magnetic stirrer (Schwabach, Germany) and the intracellular material was released by cell osmotic shock. The sample was then centrifuged at 12,000× g for 15 min (Medtronic BL-S, P-Selecta, Barcelona, Spain) to recover the phycobiliproteins solubilised in the supernatant. Subsequently, the phycobiliproteins were precipitated by adding an ammonium sulphate solution (70% saturation) and left to stand overnight in the dark at 4 °C. Following this, the purified biliprotein pellets were resuspended in a small volume of the phosphate buffer (5 mM, pH 6.5) and dialyzed overnight at 4 °C against the same buffer. Then, 1% *w*/*w* sodium azide (NaN_3_) was added to the protein solution as a preservative and the samples were stored at 4 °C until further analysis. The absorbance spectra were recorded using a UV–visible spectrophotometer (Lambda 20, Perkin Elmer, UV Data Manager software, Waltham, MA, USA) at 250–750 nm. The concentrations of phycobiliproteins (PC-Phycocyanin, APC-allophycocyanin, and PE-phycoerythrin) were estimated using the following equations:(1)$$PC\;\left(mg\cdot mL^{-1}\right)=\frac{A_{615}-0.474\times \left(A_{652}\right)}{5.34}$$
(2)$$APC\;\left(mg\cdot mL^{-1}\right)=\frac{A_{652}-0.208\times \left(A_{615}\right)}{5.09}$$
(3)$$PE\;\left(mg\cdot mL^{-1}\right)=\frac{A_{563}-2.41\times \left(PC\right)-0.849\times \left(APC\right)}{9.62}$$

### 2.3. Colour Evaluation

Colour recordings were taken using a CM-5 chroma meter (Konica Minolta, Osaka, Japan) and the CM-S100W SpectraMagic™ NX software version 2.2 (Konica Minolta, Osaka, Japan). The colouring of beverage samples was studied (Figure 1). The studied beverages were isotonic drinks, tonic water, and wine—all commercial products. The selected commercial products have a blue colour, whereas the standard products were colourless. The same brand was used for the commercial and standard products. The samples were stored in a domestic refrigerator at 5 ± 2 °C. Only the tonic drink and wine were manipulated before the colourimetric analysis to eliminate their dissolved gases. This was performed by stirring for 20 min at 300× *g*. The blue protein extract was added to the beverages while stirring constantly. The pigmentation capacity of the extract was carried out as described elsewhere [13]. 

### 2.4. Cytotoxicity

The cytotoxicity of the C-PC extracts obtained from the *Oscillatoria* sp. biomass was evaluated through a cytotoxicity assay with human gingival fibroblasts. Cells from the HGF-1 cell line (8 × 10^3^ cells well^−1^) were incubated at C-phycocyanin concentrations from 3 to 1.95·10^−2^ mg mL^−1^ in serial dilutions (1:1). These were placed in a 96-well microplate for 72 h at 37 °C with 5% CO_2_ in a humid atmosphere. Cell proliferation was estimated through the MTT (3-(4,5-dimethylthiazol-2-yl)-2,5-diphenyltetrazolium bromide) assay, as described elsewhere [14]. The relative cell viability was expressed as the percentage of viable cells compared to the control. Four samples from each tested concentration were included in each experiment. Determinations were carried out in triplicate in independent experiments.

### 2.5. Biostimulant Capacity of the Waste Biomass

Up to four different bioassays were conducted to evaluate the biostimulant properties of the *Oscillatoria* sp. extracts. For this, the leftover biomass (after extracting the C-phycocyanin) was subjected to enzymatic hydrolysis for 4 h in a stirred-tank reactor at 40 °C and pH 7.5, using Alcalase and Flavourzyme from Novozyme [15]. The gibberellin-like effect was studied by measuring the germination index of *Lepidium sativum* seeds [16]. The auxin-like effect was determined by the adventitious root induction of mung beans (*Vigna radiata*). Commercial *Vigna radiata* seeds were planted following previous work [17]. The cytokinin-like effect was determined through the excised cucumber expansion test [17]. Moreover, the antisenescent activity of the natural cytokinins was determined through the *Triticum* (Wheat) chlorophyll retention test [18]. For this, seeds from suitable wheat cultivars (*Triticum aestivum* L.) may be used. The wheat seeds were rinsed under running tap water for 4 h. Each seed was planted at a 1 cm depth in moistened perlite in glass trays. The trays were placed in a growth chamber and kept at 25 °C, at 60 to 65% relative humidity, and illuminated with fluorescent lamps for 10 days. The seedling leaves were collected and cut (3 cm below their apical tip) into 10 mm segments. The fresh weight of ten cuttings was measured with an analytical grade balance and placed in a 50 mL falcon centrifuge tube (four vials per treatment) containing 10 mL of distilled water and 0.5 mg/L BAP (as the control), and biostimulants at 0.5 and 2 g·L^−1^ (as the treatment). The vials were placed back in the dark growth chamber for 4 days. After the incubation period, the leaves were blotted dry and put into test tubes (graduated 14 mL falcons) containing 8 mL of 80% ethanol. The test tubes were transferred to a water bath (warmed to 80 °C). After 10 min of chlorophyll extraction, the solution was cooled down and supplemented with 80% ethanol up to 10 mL. To avoid evaporation, the test tubes were covered. The cooled chlorophyll extract was poured (without the segments) into spectrophotometer cuvettes. The optical density was determined at 645 nm and compared to 10 mg of fresh weight; the adjusted results were compared to the control and analysed using a standard curve for comparison made using the logarithmic concentration range of the specific cytokinin (BAP).

## 3. Results and Discussion

### 3.1. Microalgal Biomass Production

Production of the cyanobacteria *Oscillatoria* sp. in a raceway reactor (located in Almeria, Spain) was studied. This strain was selected because of its great potential for producing both C-phycocyanin and biostimulants. Moreover, the strain tolerates high salinity allowing seawater to be utilised in the production step, thus making the process more sustainable. The use of regular fertilisers as the nutrient source also reduces the biomass production cost, although this could be further reduced if nutrient-rich effluents such as wastewater and centrate (the liquid fraction of effluents from anaerobic digestion of organic matter) were used, again contributing to a more sustainable process [19]. Experiments were performed in different seasons, with the environmental conditions changing throughout the year. The solar radiation ranged from 11 to 30 MJ/m^2^·day, whereas the daily average temperature ranged from 11 to 25 °C between the winter and summer seasons (Figure 2). Due to this variation, the performance of the *Oscillatoria* cultures also varied despite operating the reactors at the same dilution rate (0.2 day^−1^) and pH (9.0). The biomass concentration ranged from 0.3 to 0.9 g/L and the biomass productivity ranged from 9.0 to 30.0 g/m^2^·day (Figure 2). The data were as expected when applying a previously validated model that considered both the biological and engineering aspects of the photobioreactor [20]. For this, the biological parameter values already considered were a maximal growth rate of 0.6 day^−1^, an irradiance constant of 120 µE/m^2^·s, and a biomass extinction coefficient of 0.12 m^2^/g. The results show that *Oscillatoria* sp. performed adequately in spring and summer, but not so in the autumn when the biomass concentration dropped below 0.5 g/L, making the risk of culture washout too high. The suitable period for producing this strain was from March to October when the average achievable biomass productivity was 21 g/m^2^·day, equivalent to 52 t/ha·year. This biomass production capacity is lower than the maximum values already reported for locations with template climate (up to 100 t/ha·year), due both to the low performance of this strain in outdoor reactors and the fact that it is not possible to produce it all-year-round. It has been reported that robust and fast-growing strains capable of being produced throughout the year are required to maximize the performance of microalgae-related processes [1]. 

### 3.2. Phycocyanins Extraction

Once the *Oscillatoria* sp. biomass was produced, different extraction processes were assayed to determine the best conditions for C-PC recovery from the microalgal biomass (Table 1). Firstly, a global biomass characterization was developed to determine the total amount of phycocyanins and chlorophylls contained in the biomass, with values of 0.5 and 0.82% (*w*/*w*) being measured, respectively. Regarding the extraction process, the influence of different variables such as the ionic strength (FI), pH, and the stirring time in the C-PC extraction were tested. The C-PC concentration, purity, and extraction ratio of the phycocyanin extracts were determined using spectroscopic measurements. In addition, different cell disruption strategies based on direct osmotic shock, freeze–thawing, ultrasound (US), and their combination have been studied.

The results show that the maximal C-PC extracted value of 0.5% *w*/*w* was obtained when combining all the cell disruption methods and performing the extraction at pH 6.5 using an FI of 100 mm; in this case, a purity ratio of 0.173 was measured (Table 1). In contrast, the highest purity ratio of 0.190 was achieved when using freeze–thawing as the cell disruption method followed by extraction at pH 6.5 and FI 100 mm. Taking into account all the extraction parameters evaluated, it seems the best procedure would be to use the freeze–thawing methodology, which generates a high protein concentration in the extract as well as high purity and extraction ratios. However, in addition to good extraction parameters, the economics of the process should also be taken into account. In this regard, direct extraction using osmotic shock could be used to obtain very similar parameters to the maximal ones. The utilization of different solvents and methods to facilitate the extraction of pigments from microalgae biomass is a highly relevant topic. In this sense, solvents such as methanol, chloroform, dimethyl sulphoxide (DMSO) and acetone have been proposed, although the use of water is preferred [21]. Concerning methods, the utilization of mechanical methods such as microwave and ultrasound to facilitate the cell breakage and extraction of molecules has been also reported, although the most recommendable is the utilization of high-pressure homogenization by its lower energy consumption and increase in temperature that can denature the target compounds [22].

The C-PC extract from *Oscillatoria* sp. was mainly composed of C-phycocyanin (C-PC). A preliminary analysis of the extract’s phycobiliprotein content revealed that the extract contained 2.08 mg C-PC·mL^−1^ as the principal component (91% *w*/*v*). Other proteins were present in lesser yet significant proportions, such as APC (9% *w*/*v*). The C-phycocyanin purity grade is defined as the ratio between its peak absorbance at 616 nm and the absorbance of proteins at 280 nm (A616/A280). When this ratio is higher than 0.7, it is considered to be of food-grade purity [23]. In this work, the C-PC purity ratio was 0.71, demonstrating that it could be used in tests as a food colouring agent without further purification. In this case, only a small proportion of APC was present, which did not significantly affect the final blue colour of the C-PC extract (APC is also a blue protein, its shade being lighter than that of C-PC). Thus, as the main extract constituent, C-phycocyanin confers an attractive, deep-dark-blue colour. The pH is a relevant factor influencing phycobiliprotein stability. The beverages evaluated in this work had a pH of 3.0 (isotonic drink and tonic water) and a pH of 3.2 (wine). To obtain a stable medium for the C-PC, the pH of the beverages was corrected to 5.0, a level at which the extract’s blue colour is stable. 

### 3.3. Utilization of the C-PC Extract as a Colourant 

To determine the feasibility of using the already obtained C-PC extract as a colouring agent, a set of experiments were performed using different soft drinks. The colour exhibited by the blue C-PC extract obtained in this work was compared with the colour of the reference commercial products that were coloured using synthetic molecules. The most common synthetic blue food colours are anthraquinone blue (E-130), blue patent V (E-131), indigotine (E-132), and brilliant blue (E-133) while the most frequent natural molecule is anthocyanin (E-163) [24]. The standard commercial product was pigmented using the blue C-PC extract to achieve a colour that was as close as possible to that of the reference products. For this, it was necessary to add a specific volume of C-PC extract to the uncoloured commercial beverages (Table 2). It should be noted that the effect of the extract addition on texture and viscosity was negligible given that the added quantities needed to reproduce the colours of the corresponding market products were lower than 106 mg·L^−1^ in all the tests. Similar results were reported in previous works where natural pigments did not affect the rheological properties of dairy products [25]. In this work, tonic beverages required a smaller amount of C-PC to match the colour of the commercial products. Samples D and E had staining factors of 22 and 47.9 mg·L^−1^, respectively. In contrast, Sample B was the product requiring the most protein, having a staining factor of 105.8 mg·L^−1^. These results show the C-PC extract’s potential as a natural colourant while the low staining factor values confirm its economic viability. Figure 1 shows the samples coloured with the C-PC extract compared with the reference products found on the market. These images demonstrate that the colour of each sample was similar to the colour of the commercial reference product. In addition, Figure 3 shows the colouring curves of the different beverages. When the ΔE*ab value is in the 0.0–0.5 range, the colour difference is imperceptible to the human eye, whereas values in the 0.5–1.5 range are slightly different and higher values are easily detected by an untrained consumer [26].

Three different commercial isotonic beverages were studied to test the potential colouring capacity of the C-PC extract (Table 2). The final colours present notable differences, although these are probably not enough for consumers to find them unattractive. Further studies will assess consumer acceptance of the products. The products developed in this work had the additional advantage of being produced using a natural colourant. Recent work demonstrated that, if consumers are aware of the health and environmental benefits of incorporating natural microalgae-derived products into foods, not only their purchase intention but also the amount of money they are willing to pay for the product is greater [27]. Sample B registered the highest staining factor (Table 2) and its colouring curve suggests that a large volume of the B-PE extract (750 μL) was needed to achieve the final sample colour (Figure 3). However, in all the studied samples, the staining factors were low, proving that attractive colours can be achieved with a very low quantity of the C-PC. 

Cytotoxicity tests were performed to evaluate the safety of the produced extracts to be used as colourants. The results show that there was C-PC extract cytotoxicity up to a concentration of 196.87 µg mL^−1^, with a survival rate of 38%. From this concentration, the highest survival rate was 80% at a concentration of 98.43 µg mL^−1^ (Figure 4A). The IC_50_ obtained was 169.86 µg mL^−1^. Additionally, a cytotoxicity analysis was performed on Sigma’s pure phycocyanin (99.9%) to see if the cytotoxicity was due to the C-PC itself or to other extract proteins or components. The analysis showed that high-purity C-PC is not toxic to the HFG-1 cell line (Figure 4B). Based on these figures, it can be concluded that some compounds already contained in the extract from algae biomass can be slightly toxic. As the concentration range at which cytotoxicity was observed was much higher than the dose needed to provide adequate colour, it was concluded that the colouring extracts could be used as colourants. However, more research is necessary to identify and remove the other compounds accompanying the C-PC in the extracts, which may be responsible for toxicity in the samples at high concentrations.

### 3.4. Stability of Coloured Beverages 

The stability of food products is a pivotal aspect to consider; so, colour stability determinations were developed to test the beverages to which the C-PC extract was added. Figure shows the colour changes of the products during an 11-day storage period. Regarding the isotonic and tonic samples, very slight variations were observed during the storage period. In this case, the tonic sample showed the highest a* values while the isotonic sample registered the lowest. Similarly, the L* (Figure 5B) and b* (Figure 5D) values for the isotonic and tonic samples remained constant and stable from day 0 to day 11. Despite variations and upward trends, the ΔE*ab did not exceed a value of 0.75 over the entire study period for the tonic and isotonic samples. A different trend was observed for the wine sample, which started with a value of ΔE*ab = 0.8 on day 1 and concluded with ΔE*ab = 2.6 on day 11. It is well-known that ΔE*ab values below 3.0 cannot be easily detected by the naked human eye and are interpreted as the same colour by consumers [28]. According to these data, it could be concluded that the colour was stable in all the samples assayed over the 11 days, the stability being higher in the isotonic and tonic samples. Concerning the stability studies of the beverage samples, the data obtained proved that the colour was stable in the isotonic and tonic samples. This effect might be explained by the antioxidant activity of the phycobiliproteins [29,30]. Variations of the stability of colour between the different samples are related to the composition of the matrices—especially the pH and ionic strength—that can denature the phycobiliproteins and then modify the colour of the samples along the time. 

### 3.5. Biostimulant Activity of the Residual Biomass

It is proposed that, once the C-PC extract has been obtained from the *Oscillatoria* sp. biomass, the waste biomass is used as a biostimulant for plant growth. Due to the various possible biostimulant effects, there is no universal protocol that allows one to predict the biostimulant capacity of a natural extract. In this work, four bioassays were carried out.

Gibberellins are a group of plant hormones that play an important role in initiating seed germination and stem elongation. The gibberellin-like effect promoted by the microalgae was observed through the germination index (GI) of watercress seeds. In this work, a GI value of 100% was attributed to the control (distilled water). As shown in Figure 6A, samples with microalgae added resulted in values above 100%, which is considered to be due to their biostimulant activity. The highest GI was achieved with a microalgal biomass concentration of 0.5 g·L^−1^, which led to a GI of 123% ± 13%, although even the 0.1 g·L^−1^ solution improved the GI. This demonstrates the presence of molecules that have biological activity, such as gibberellins, in the microalgal biomass. Auxins play a crucial role in the induction of root initiation and elongation growth; so, the existence of auxin-like activity of the microalgal biomass was also studied. As shown in Figure 6B, the 2 g·L^−1^ biomass sample led to a greater number of roots in the mung bean cotyledons compared with the control (distilled water), thus proving the presence of auxins. Although microalgae cannot promote mung bean root development as much as the direct addition of synthetic hormones, they did achieve a high value of 168% ± 84%. Concerning the cytokinins, these are a group of phytohormones that are supposed to control cell division, bud development, and leaf blade development. The cytokinin-like effect of the microalgal biomass was checked using the cucumber cotyledon root expansion test. As shown in Figure 6C, adding the *Oscillatoria* biomass solutions at 0.5 and 2 g·L^−1^ led to a 190% ± 62% and 175% ± 46% expansion, respectively, much greater than the distilled water control sample although lower than when using synthetic hormones. In addition, the chlorophyll retention test was performed to determine the cytokinin-like activity of the microalgal biomass. This was conducted on wheat seeds instead of cucumber seeds. Here, adding synthetic hormones (BAP 5 mg/L) led to a significant increase in chlorophyll retention, up to 270% ± 36%. In comparison, adding microalgal biomass slightly enhanced chlorophyll retention, reaching values of 108% ±28% when using 0.5 g·L^−1^ and 122% ±28% when 2 g·L^−1^ of microalgal biomass was used.

The results confirm the potential use of leftover *Oscillatoria* biomass as a biostimulant product since all the bioassays carried out in this work demonstrated the presence of gibberellin-like (the germination index), auxin-like (mung bean root formation), and cytokinin-like (cucumber expansion and chlorophyll retention) effects. Similar results were found using different microalgae and cyanobacteria such as *Scenedesmus obliquus* [31], *Nostoc piscinale* [32], and *Chlorella* [33,34], amongst others. The benefits of using these microalgae-related biostimulants in agriculture have been widely reported, although more research is still needed to identify the mechanisms involved in improved crop performance [8,12].

### 3.6. Techno-Economic Analysis of the Proposed Biorefinery

These results confirm that *Oscillatoria* sp. can be produced outdoors in open reactors using seawater and fertilizers as the culture medium. Moreover, the biomass produced contains C-PC, which can be extracted for use as a safe food colourant, with the residual biomass also being useful as a biostimulant for agriculture-related applications. Once the technical feasibility of the process and products was demonstrated, the data could be used to perform a preliminary economic analysis to identify the most relevant bottlenecks. Based on previously obtained knowledge regarding microalgae production costs and processing costs for different products, a base case study was defined for a unitary surface of 1 ha. In this case study, the experimentally determined biomass production capacity and C-PC content were used, in addition to the amount of biostimulant obtained per mass unit of microalgae biomass (Table 3). Concerning the biomass production cost, based on previous studies, a unitary biomass production cost of 5 €/kg was established, whereas for the extraction of the C-PC and biostimulants, unitary costs of 30 €/kg and 2 €/kg were set, respectively [1]. Regarding the value of the products, a biomass value of 10 €/kg was considered, similar to that reported for *Spirulina*, whereas for the C-PC and biostimulant, values of 500 €/kg and 5 €/kg were set, respectively [1]. Based on these values, the facility’s turnover in different scenarios was calculated (Figure 7). The results show that the overall economic balance is always positive, except when C-PC extracts are considered as a single product. In the latter case, the turnover is negative, even when a +20% variation in the product value is considered. The value of the C-PC extract would have to be at least 1000 €/kg to achieve a neutral turnover; otherwise, the C-PC content in the biomass would have to double, above 1.0% d.wt. This second scenario would seem more plausible. Concerning the biomass production, the turnover is positive because the biomass value is double that of the production cost; even when considering a 20% decrease in the biomass value, the turnover is always positive. Concerning the production of biostimulants, this scenario shows the largest turnover for single use of the biomass, as the biostimulant is much more valuable than the biomass in which it is contained. 

In terms of the biorefinery schemes, producing C-PC extracts while valorising the residual biomass is profitable, although the best scenario is one in which both C-PC extracts and biostimulants are produced (Figure 7). These results confirm the benefits of developing biorefineries capable of increasing the number and value of the products already obtained from the biomass, which results in increased profits from the process. However, not all the scenarios are positive as the selection of suitable products is critical. The findings indicate that high-value products are not always the most suitable to obtain, especially if the valuable compounds are present in the biomass at concentrations that are too low. Instead, producing large amounts of medium-value products is very important when it comes to maximising the benefits of microalgae-related processes.

## 4. Conclusions

The production of *Oscillatoria* sp. in outdoor reactors using seawater and fertilisers is possible, although the production capacity must be increased to achieve maximal values. The produced biomass contains valuable compounds such as C-PC and biostimulants. The C-PC extracts were shown to be safe for use as a food colourant; furthermore, the colourant is stable and only a low dosage is required to obtain colours close to the commercial soft drink and wine products. The residual biomass was also demonstrated to be an effective biostimulant, promoting seed germination and plant growth. A preliminary economic analysis showed that, if producing only a single product, biostimulants are the most interesting scenario. However, the best scenario of all is to develop biorefineries capable of producing both C-PC extracts (as food colourants) and biostimulants. Producing only C-PC extracts as the final product is not recommended. The results emphasize the importance of carrying out an in-depth analysis of the products to be obtained before designing any microalgae-related process.

## Figures and Tables

**Figure 1 biology-11-01278-f001:**
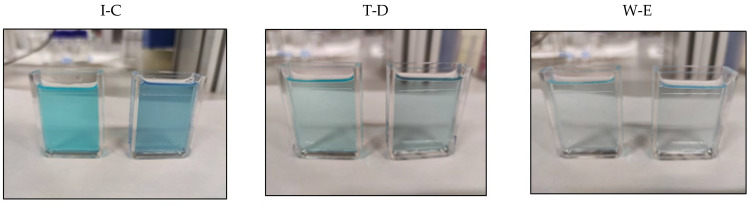
Photographs showing a vivid presentation of the commercial beverages (on the left of each pair) and the beverages containing the C-PC extract as a natural colourant (on the right of each pair). Three representative samples are shown: I-C representing isotonic beverages, T-D representing tonic beverages, and W-E representing wine.

**Figure 2 biology-11-01278-f002:**
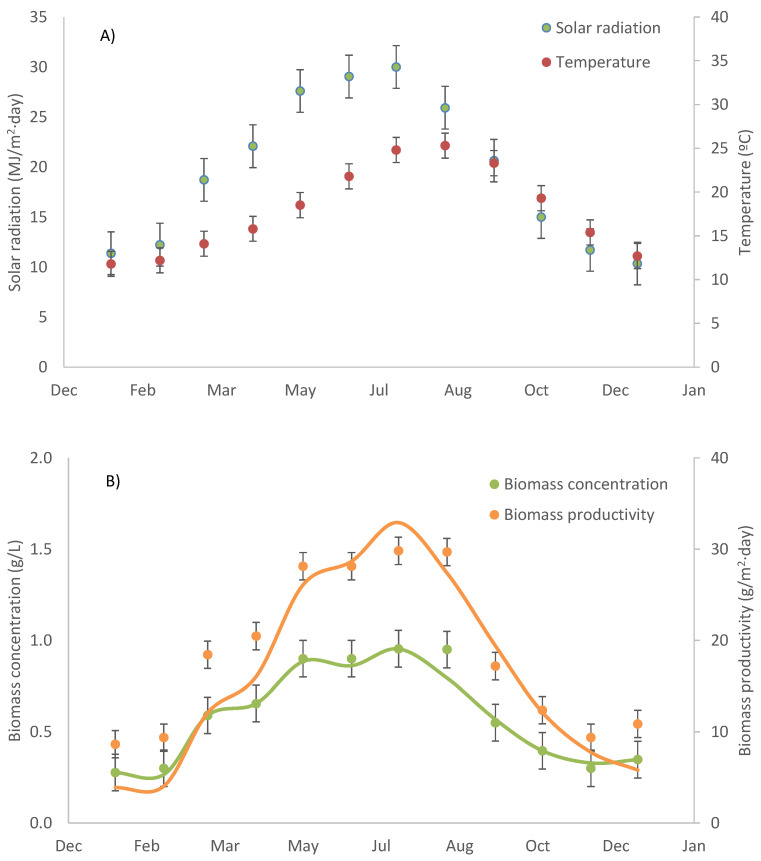
Annual variation in (**A**) environmental conditions (solar radiation and culture temperature) and (**B**) biomass production (concentration and productivity) of *Oscillatoria* sp. cultures cultivated in an 80 m^2^ raceway reactor operated in semicontinuous mode at 0.2 day^−1^. Lines correspond to values estimated using the previously developed model [20].

**Figure 3 biology-11-01278-f003:**
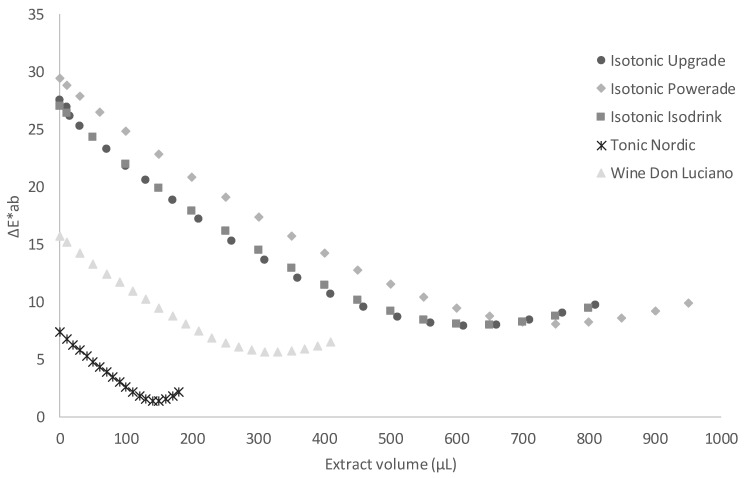
Colouring curves of beverages obtained from colouring assays using the C-PC extract from *Oscillatoria* sp. ([C-PC] extract = 2.08 mg mL^−1^).

**Figure 4 biology-11-01278-f004:**
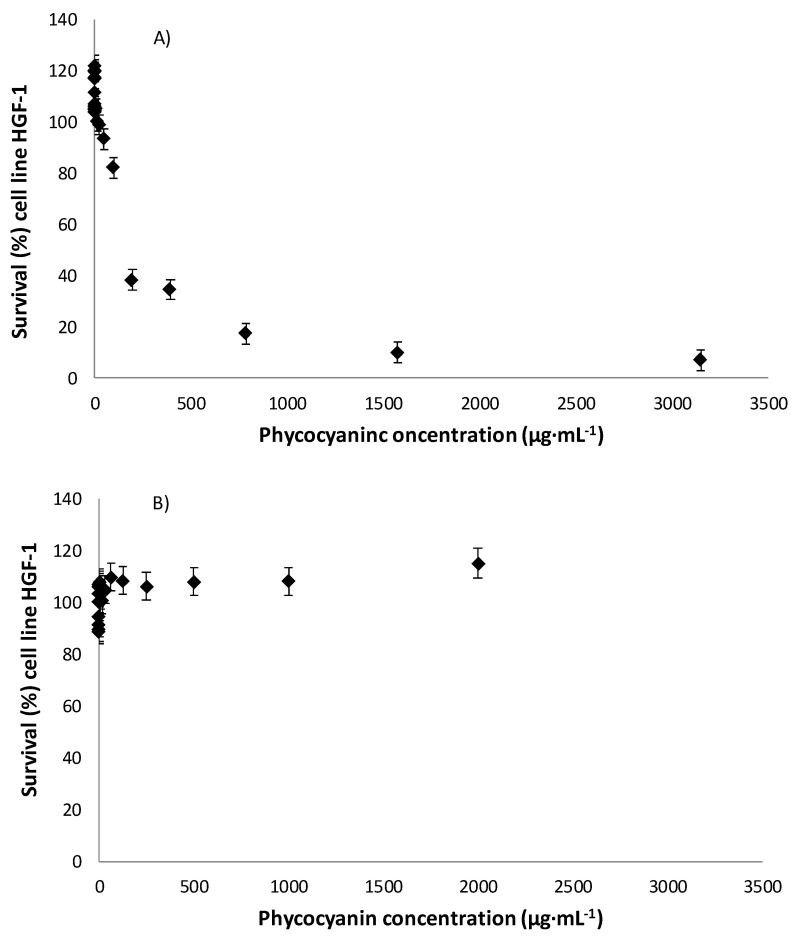
Survival of the HGF-1 cell line as a function of the C-phycocyanin concentration for the obtained extract (**A**) and Sigma’s 99.9% purity of phycocyanin (**B**).

**Figure 5 biology-11-01278-f005:**
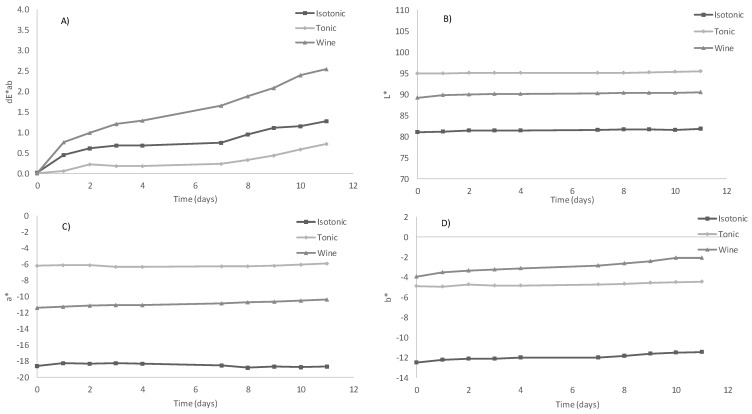
Changes in the colour indexes of the isotonic beverage, the tonic beverage, and the wine pigmented with the C-PC extract during a short cold-storage period. The plotted values are the means of the triplicate tests. Graphs show changes in ∆E*ab (**A**), L* (**B**), a* (**C**), and b* (**D**).

**Figure 6 biology-11-01278-f006:**
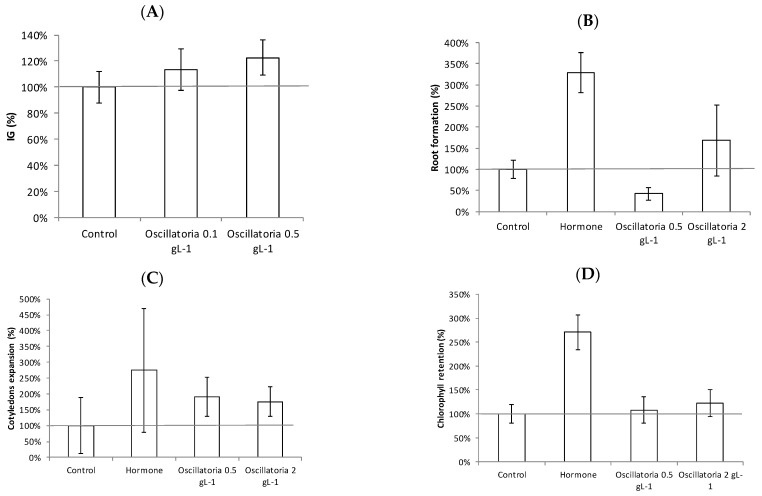
Germination index results: gibberellin-like effect in watercress seeds (**A**). Root formation results: auxin-like effect in mung beans (**B**). Cotyledon expansion results: cytokinin-like effect in cucumber seeds (**C**). Chlorophyll retention results: cytokinin-like effect in wheat seeds (**D**).

**Figure 7 biology-11-01278-f007:**
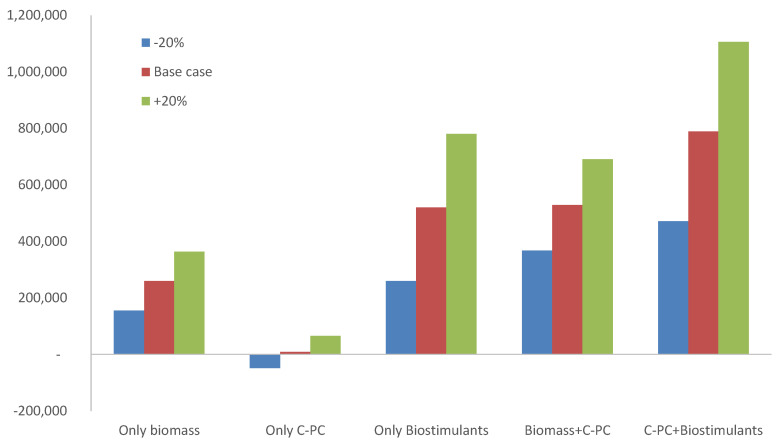
Economic analysis of different scenarios for a 1-ha facility producing *Oscillatoria* sp. for different purposes (biomass, C-PC extracts used as a food colourant, and biostimulants).

**Table 1 biology-11-01278-t001:** Extraction processes assayed for C-phycocyanin recovery (US stands for ultrasound treatment).

Method	pH	FI (mm)	C-PC (mg/mL)	Purity Ratio (A_max_/A_280_)	C-PC Extracted (% *w*/*w*)
Direct (5 min stirring)	6.5	5	0.23	0.146	0.33
Freeze–thawing	6.5	5	0.18	0.100	0.26
Freeze–thawing	6.5	50	0.34	0.150	0.37
Freeze–thawing	6.5	100	0.42	0.190	0.45
Freeze–thawing	6.5	175	0.43	0.180	0.46
Freeze–thawing	6.5	250	0.30	0.160	0.49
US + Freeze–thawing	7.0	100	0.34	0.182	0.49
US + Freeze–thawing	6.5	100	0.34	0.187	0.48
US + Freeze–thawing	5.5	100	0.33	0.180	0.47
US + Freeze–thawing	5.0	100	0.33	0.170	0.46
US + Stirring 45 min + Freeze–thawing	6.5	100	0.32	0.173	0.50

Biomass characterization: C-PC = 0.5% d.wt. biomass; Chlorophylls = 0.82% d.wt. biomass (0.67%chlA; 0.15% chlC).

**Table 2 biology-11-01278-t002:** Staining factors of the assayed beverages: data are obtained from the colouring curves of the products ([C-PC] extract = 2.08 mg/mL^−1^).

Product Type	Commercial Brand	Extract Volume Added (μL)	∆E*ab Reached	Staining Factor (mg·L^−1^)
Isotonic Drinks	A	610	7.92	86.4
	B	750	8.06	105.8
	C	650	7.94	92.3
Tonic Drink	D	150	1.41	22.0
Wine	E	130	5.62	47.9

**Table 3 biology-11-01278-t003:** Values for the base case study of a 1-ha facility producing *Oscillatoria* sp. biomass in open reactors for the production of biomass, C-PC extracts, and/or biostimulants.

Base Case	
Production scale, ha	1.00
Biomass productivity, t/ha·year	52.00
C-PC content, % d.wt.	0.50
Biostimulant ratio, L/kg	5.00
Biomass production, kg/year	52,000.00
Biomass production cost, €/kg	5.00
Biomass production cost, €/year	260,000.00
C-PC production, kg/year	260.00
C-PC extraction cost, €/kg	30.00
C-PC extraction cost, €/year	7800.00
Biostimulant production, kg/year	260,000.00
Biostimulant production cost, €/kg	2.00
Biostimulant production cost, €/year	520,000.00
Biomass value, €/kg	10.00
C-PC value, €/kg	500.00
Biostimulant value, €/kg	5.00

## Data Availability

Not applicable.

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
