# Peer review of "Biorefinery Approach Applied to the Production of Food Colourants and Biostimulants from Oscillatoria sp."

_biology, 2022, doi:10.3390/biology11091278_

Round 1
Reviewer 1 Report
The overall manuscript is well written and the topic is interesting so the novelty of the research is accomplished. However, minor changes would enhance not the quality of the research, which it is already accomplished but its format. Below, I detail my suggestions:
Title:
Line 2, Biorefinery for the production of food colourants and biostimulants from Oscillatoria sp.
Biorefinery is a broad term. You may use another word next to the word biorefinery, for instance approach - Biorefinery approach applied to the production of food colorants and biostimulants from Oscillatoria sp.
Abstract
Line 14, “In this paper” change to “in this study?
Line 17, how many open raceway reactors are there?
Line 19, You are using abbreviation IF, however you have not defined this yet in the manuscript. It needs to be defines when first appears in the manuscript.
Line 24, 12 days seem to be a short period, you may say just 12 days.
Line 29, “biorefinery” is a term that defines a facility. What you have done describes a procedure for Oscillatoria sp. processing for colorant extraction and valorization of the microalgal biomass residue, thus you have used the concept or the approach.
Line 53, the term used for these activity is not “extracted” but “recovered”, since treatments other than extractions can be used for reutilization of residual biomass after high-value added compound as such as pigments, and so on.
Line 65, include reference
Line 71, include reference
Line 89, residues? Residual Oscillatoria microalgal biomass?
Line 94 -96, write the compound formulas and units with subindexes.
Line 147, a standardized protocol for cytotoxicity was used? Or was it developed by this team group?
Line 157, The word “waste” should be avoid in the context of a biorefinery approach, since this is an integrated concept in which each residue may be a feedstock for another process for the recovery of other chemical compound.
Line 182, the word “segments is referrig to a precipitate or to some residual biomass?
Line 188, include the word “Microalgal”
Line 192, the word “seawater” is to broad, you may need to narrow it to the type or conditions you are referring to.
Line 195 “wastewater and centrate were used…” . What is centrate?
Line 348, change word “waste” by word “residual”
Author Response
Reviewer 1
The overall manuscript is well written and the topic is interesting so the novelty of the research is accomplished. However, minor changes would enhance not the quality of the research, which it is already accomplished but its format. Below, I detail my suggestions:
Response: We appreciate the positive comment from the reviewer.
Title:
Line 2, Biorefinery for the production of food colourants and biostimulants from Oscillatoria sp.
Biorefinery is a broad term. You may use another word next to the word biorefinery, for instance approach - Biorefinery approach applied to the production of food colorants and biostimulants from Oscillatoria sp.
Response: agree with the reviewer the title has been modified.
Abstract
Line 14, “In this paper” change to “in this study?
Response: the proposed modification has been performed.
Line 17, how many open raceway reactors are there?
Response: the proposed modification has been performed.
Line 19, You are using abbreviation IF, however you have not defined this yet in the manuscript. It needs to be defines when first appears in the manuscript.
Response: the proposed modification has been performed.
Line 24, 12 days seem to be a short period, you may say just 12 days.
Response: the proposed modification has been performed.
Line 29, “biorefinery” is a term that defines a facility. What you have done describes a procedure for Oscillatoria sp. processing for colorant extraction and valorization of the microalgal biomass residue, thus you have used the concept or the approach.
Response: the proposed modification has been performed.
Line 53, the term used for these activity is not “extracted” but “recovered”, since treatments other than extractions can be used for reutilization of residual biomass after high-value added compound as such as pigments, and so on.
Response: the proposed modification has been performed.
Line 65, include reference
Response: the reference (Markou and Nerantzis, 2013) was already included.
Line 71, include reference
Response: the necessity of toxicity tests to approve the utilization of whatever compound in the food matrix is a legal requirement.
Line 89, residues? Residual Oscillatoria microalgal biomass?
Response: The word “residues” is not included in this paragraph.
Line 94 -96, write the compound formulas and units with subindexes.
Response: The formulas and units have been revised.
Line 147, a standardized protocol for cytotoxicity was used? Or was it developed by this team group?
Response: The protocol for cytotoxicity tests was developed by the research group but it is based on standard protocols. On the paper, only the description of the protocol is included avoiding to use of the term “standard” because it also has implications in terms of “certification” of the laboratory and whatever.
Line 157, The word “waste” should be avoid in the context of a biorefinery approach, since this is an integrated concept in which each residue may be a feedstock for another process for the recovery of other chemical compound.
Response. Agree with the reviewer the term “waste” or “residual” has been replaced by “leftover”.
Line 182, the word “segments is referrig to a precipitate or to some residual biomass?
Response: the word “segments” is used to indicate the fraction of the plants already utilized for bioassays about biostimulant activity. It is not referred to as the microalgae biomass, but as the plants utilized on these bioassays.
Line 188, include the word “Microalgal”
Response: Agree with the reviewer this modification was done.”
Line 192, the word “seawater” is to broad, you may need to narrow it to the type or conditions you are referring to.
Response: seawater is a general term used to define marine water containing a narrow range of salinity of 30 g/L.
Line 195 “wastewater and centrate were used…” . What is centrate?
Response: Centrate is the liquid phase from anaerobic digestion of effluents rich in organic matter.
Line 348, change word “waste” by word “residual”
Response. Agree with the reviewer the term “waste” or “residual” has been replaced by “leftover”.
Reviewer 2 Report
Manuscript ID: Biology-1862872
Manuscript Title: Biorefinery for the production of food colourants and biostimulants from Oscillatoria sp
Reviewer Comments
The manuscript is well-written. The authors have done a very comprehensive evaluation of the algal growth and its subsequent processing to obtain phycocyanin and the use of residual biomass as biostimulants. The work reflects the algal biorefinery concept and is very interesting for scholars working with microalgae. However, the following corrections must be implemented before the manuscript is accepted for publication.
1. Pg. No. 3, Section 2.1. What was the pH of the culture broth during the phase of cultivation?
2. Abbreviate PC, APC and PE in Equations 1, 2 and 3 mentioned in Pg. 3.
3. Pg. 5, 211-212. What do the authors mean by template locations? Also, rephrase the statement to make it readable.
4. Fig. 2: The data corresponds to air or water temperature. Authors need to cite reference and discuss
Section 3.2: Please refer and discuss the paper https://doi.org/10.1016/j.biteb.2021.100775 for the use of mechanical methods in pigment extraction.
6. Line 261, Pg. 8, How do the authors confirm that the extracts have a stable colour at pH 5.
7. What was the reason of adding different extract volume to different products shown in Table 2. Also, authors must mention the range of staining factor needed for any beverage to be acceptable in the market.
8. Figure 4: Authors must cite the reason for different patterns of survival rate obtained for the cell line for commercial phycocyanin and the extracts from algae.
9. Figure 5A: Authors must explain why for isotonic beverages, the pattern is different compared to tonic and wine beverages in section 3.3. Also, in section 3.3, why was the time-period of 11 days selected for the colour stability study.
1 Section 3.5: Authors must refer and cite: https://doi.org/10.1016/j.indcrop.2020.112453
1 Table 3: Authors can avoid mentioning the units with the numeric values provided on the right-hand side column as it seems repetitive. Also, if possible try to justify the economics including the payback time and return on investment for each scenario.
1 Section 2.1: Authors have mentioned that biomass is produced in semi-continuous mode with a dilution rate of 0.3 d-1, however, figure 2 description tells it to be in continuous mode. Authors must clarify the ambiguity.
1 Literature data in the Introduction section must be enhanced. Please cite references.
1 Kindly avoid using “our”, or “we” in the manuscript. Please use the third person/passive voice for writing.
1 Also, strengthen the novelty aspects in the last section of the introduction as well as in the abstract and conclusion.
1 Check for grammatical and typographical errors throughout the manuscript. Also, the species name should be italicized.
Author Response
Reviewer 2
The manuscript is well-written. The authors have done a very comprehensive evaluation of the algal growth and its subsequent processing to obtain phycocyanin and the use of residual biomass as biostimulants. The work reflects the algal biorefinery concept and is very interesting for scholars working with microalgae. However, the following corrections must be implemented before the manuscript is accepted for publication.
Response: We appreciate the positive comments from the reviewers.
- Pg. No. 3, Section 2.1. What was the pH of the culture broth during the phase of cultivation?
Response: the pH was set at 8.0 to minimize carbon losses. A sentence has been included to clarify that both pH and dissolved oxygen were controlled.
- Abbreviate PC, APC and PE in Equations 1, 2 and 3 mentioned in Pg. 3.
Response: Agree with the reviewer the abbreviations have been detailed.
- Pg. 5, 211-212. What do the authors mean by template locations? Also, rephrase the statement to make it readable.
Response: the sentence has been rewritten to indicate the locations corresponding to template climate (moiderate light and temperature).
- Fig. 2: The data corresponds to air or water temperature. Authors need to cite reference and discuss
Reference: data corresponds to the temperature of the water. It has been indicated in the figure caption. These values are already discussed in section 3.1 to justify the variation of biomass productivity along the year.
Section 3.2: Please refer and discuss the paper https://doi.org/10.1016/j.biteb.2021.100775 for the use of mechanical methods in pigment extraction.
Response: agree with the reviewer this paper and others have been referenced to discuss the relevance of solvents and extraction methods applied.
- Line 261, Pg. 8, How do the authors confirm that the extracts have a stable colour at pH 5.
Response: stability of colour was evaluated by measurements of the colour of the samples for two weeks.
- What was the reason of adding different extract volume to different products shown in Table 2. Also, authors must mention the range of staining factor needed for any beverage to be acceptable in the market.
Response: the volume of extract added to the different matrices was defined to achieve a colour close to the commercial products. There is not a fixed staining factor for each product. The challenge is to find the adequate concentration of pigment required to achieve the commercial colour. This sentence was already included into the text “The standard commercial product was pigmented using the blue C-PC extract to achieve a colour that was as close as possible to that of the reference products.”
- Figure 4: Authors must cite the reason for different patterns of survival rate obtained for the cell line for commercial phycocyanin and the extracts from algae.
Response: Agree with the reviewer the next sentence has been included: “Based on these figures it can be concluded that some compounds already contained in the extract from algae biomass can be slightly toxic”.
- Figure 5A: Authors must explain why for isotonic beverages, the pattern is different compared to tonic and wine beverages in section 3.3. Also, in section 3.3, why was the time-period of 11 days selected for the colour stability study.
Response: Agree with the reviewer the next sentence has been included: “Variations of the stability of colour between the different samples are related to the composition of the matrices, especially the pH and ionic strength, that can denature the phycobiliproteins and then modify the colour of the samples along the time.”
1 Section 3.5: Authors must refer and cite: https://doi.org/10.1016/j.indcrop.2020.112453
Response: Agree with the reviewer this paper has been included into the references list.
1 Table 3: Authors can avoid mentioning the units with the numeric values provided on the right-hand side column as it seems repetitive. Also, if possible try to justify the economics including the payback time and return on investment for each scenario.
Response: agree with the reviewer the units have been removed from the data column. Because this is a preliminary economic analysis no further calculations about payback, benefits or other indexes were calculated. To do that we will need more robust data from large-scale facilities confirming all the costs and assumptions already included in this preliminary analysis.
1 Section 2.1: Authors have mentioned that biomass is produced in semi-continuous mode with a dilution rate of 0.3 d-1, however, figure 2 description tells it to be in continuous mode. Authors must clarify the ambiguity.
Response: the caption in figure 2 has been modified to include the word “semicontinuous” instead of continuous. The production is performed daily harvesting a certain volñume of the culture, but it is not performed along the 24 h, but in 4 hours in the morning, then it is more appropriate to use the term “semicontinuous”.
1 Literature data in the Introduction section must be enhanced. Please cite references.
Response: the introduction section contains relevant references and data about thew topic of the study.
1 Kindly avoid using “our”, or “we” in the manuscript. Please use the third person/passive voice for writing.
Response: agree with the reviewers these terms have been removed.
1 Also, strengthen the novelty aspects in the last section of the introduction as well as in the abstract and conclusion.
Response: We consider that the relevance of data and phenomena already included in the paper has been exposed in these sections.
1 Check for grammatical and typographical errors throughout the manuscript. Also, the species name should be italicized.
Response: The entire manuscript has been revised.
Reviewer 3 Report
Paragraph 2.2: extraction and characterization of phycocyanin
Extraction of 5 min is very short. There is no kinetic of phycocyanin concentration in crude extract to claim that 100% of C-PC was extracted during this period. Do the authors have data (not published or reference) to claim that 5 min is sufficient to extract all C-PC?
Equations 1, 2 and 3 are from “Bennett & Bogorad 1973, Complementary chromatic adaptation in a filamentous blue-green alga, The Journal of Cell Biology”. Reference is missing.
C-PC purity ratio (Amax/A280) is not described in 2.2
Author Response
Paragraph 2.2: extraction and characterization of phycocyanin
Extraction of 5 min is very short. There is no kinetic of phycocyanin concentration in crude extract to claim that 100% of C-PC was extracted during this period. Do the authors have data (not published or reference) to claim that 5 min is sufficient to extract all C-PC?
Reference: in previous studies, we confirm that extraction for 5 min allows recovery of more than 80% of the C-PC contained in the biomass. The extraction method can be optimized in further strudies but for this work, it was not critical.
Equations 1, 2 and 3 are from “Bennett & Bogorad 1973, Complementary chromatic adaptation in a filamentous blue-green alga, The Journal of Cell Biology”. Reference is missing.
Response: this reference is the right one, we use it for a long time, but in the last few times some reviewers critics to use of old references indicating that it is well-established knowledge.
C-PC purity ratio (Amax/A280) is not described in 2.2
Response: It was considered not necessary to include an equation to calculate this ratio because it is very easily calculated from experimental measurements.
Round 2
Reviewer 3 Report
OK